# Clinical academic research in the time of Corona: A simulation study in England and a call for action

Amitava Banerjee[1,2,3]*, Michail Katsoulis[2,3], Alvina G. Lai[2,3], Laura Pasea[2,3], Thomas A. Treibel[4], Charlotte Manisty[4], Spiros Denaxas[2,3], Giovanni Quarta[5], Harry Hemingway[2,3], João L. Cavalcante[6], Mahdad Noursadeghi[7], James C. Moon[1,4]

1 Barts NHS Trust, London, United Kingdom, 2 Health Data Research UK, University College London, London, United Kingdom, 3 Institute of Health Informatics, University College London, London, United Kingdom, 4 Institute of Cardiovascular Science, University College London, London, United Kingdom, 5 Department of Cardiology, Ospedale Papa Giovanni XXIII, Bergamo, Italy, 6 Minneapolis Heart Institute, Minneapolis, Minnesota, United States America, 7 Division of Infection and Immunity, University College London, London, United Kingdom

* ami.banerjee@ucl.ac.uk

**Data Availability Statement:** All data are available as supplementary files.

## Abstract

### Objectives

We aimed to model the impact of coronavirus (COVID-19) on the clinical academic response in England, and to provide recommendations for COVID-related research.

### Design

A stochastic model to determine clinical academic capacity in England, incorporating the following key factors which affect the ability to conduct research in the COVID-19 climate: (i) infection growth rate and population infection rate (from UK COVID-19 statistics and WHO); (ii) strain on the healthcare system (from published model); and (iii) availability of clinical academic staff with appropriate skillsets affected by frontline clinical activity and sickness (from UK statistics).

### Setting

Clinical academics in primary and secondary care in England.

### Participants

Equivalent of 3200 full-time clinical academics in England.

### Interventions

Four policy approaches to COVID-19 with differing population infection rates: "Italy model" (6%), "mitigation" (10%), "relaxed mitigation" (40%) and "do-nothing" (80%) scenarios. Low and high strain on the health system (no clinical academics able to do research at 10% and 5% infection rate, respectively.

**Funding:** AB is supported by research funding from NIHR, British Medical Association, Astra-Zeneca and UK Research and Innovation. HH is a National Institute for Health Research (NIHR) Senior Investigator and funded by the National Institute for Health Research University College London Hospitals Biomedical Research Centre. HH work is supported by: 1. Health Data Research UK (grant No. LOND1), which is funded by the UK Medical Research Council, Engineering and Physical Sciences Research Council, Economic and Social Research Council, Department of Health and Social Care (England), Chief Scientist Office of the Scottish Government Health and Social Care Directorates, Health and Social Care Research and Development Division (Welsh Government), Public Health Agency (Northern Ireland), British Heart Foundation and Wellcome Trust. 2. The BigData@Heart Consortium, funded by the Innovative Medicines Initiative-2 Joint Undertaking under grant agreement No. 116074. This Joint Undertaking receives support from the European Union's Horizon 2020 research and innovation programme and EFPIA; it is chaired, by DE Grobbee and SD Anker, partnering with 20 academic and industry partners and ESC. The funders had no role in study design, data collection and analysis, decision to publish, or preparation of the manuscript.

**Competing interests:** No competing interests for any authors.

## Main outcome measures

Number of full-time clinical academics available to conduct clinical research during the pandemic in England.

## Results

In the "Italy model", "mitigation", "relaxed mitigation" and "do-nothing" scenarios, from 5 March 2020 the duration (days) and peak infection rates (%) are 95(2.4%), 115(2.5%), 240 (5.3%) and 240(16.7%) respectively. Near complete attrition of academia (87% reduction, <400 clinical academics) occurs 35 days after pandemic start for 11, 34, 62, 76 days respectively—with no clinical academics at all for 37 days in the "do-nothing" scenario. Restoration of normal academic workforce (80% of normal capacity) takes 11, 12, 30 and 26 weeks respectively.

## Conclusions

Pandemic COVID-19 crushes the science needed at system level. National policies mitigate, but the academic community needs to adapt. We highlight six key strategies: radical prioritisation (eg 3–4 research ideas per institution), deep resourcing, non-standard leadership (repurposing of key non-frontline teams), rationalisation (profoundly simple approaches), careful site selection (eg protected sites with large academic backup) and complete suspension of academic competition with collaborative approaches.

## Introduction

The pandemic SARS-CoV-2 virus (causing the disease, COVID-19) is unprecedented in its impact on individuals, populations and health systems [1]. Since the first cases in Wuhan, China in November 2019, every country has been affected [2,3] but with wide variations in the ability and capacity to respond, with only half estimated to have operational readiness [4]. Countries hit later can benefit and learn from acquired knowledge and experience of preceding countries. As part of this response, the research effort is crucial for development, testing and adoption of effective preventative and treatment [5,6].

A Pubmed search (November 2019 to April 2020), using terms "coronavirus" and "COVID-19" showed 2206 and 1604 articles respectively, suggesting swift global research mobilisation. However, the publication mix shows the vast majority are reviews, opinions and commentary rather than formal research. Many publications on COVID-19 are not clinically led, and many are not directly clinically informed. "Learning is difficult in the midst of an emergency" [7], but our ability to deliver timely, high-impact clinical research, relevant to patients and populations, is critical across the academic spectrum [8], from "bench to bedside to big data", whether basic biology, repurposed and novel therapeutic approaches, vaccines or modelling. Obstacles to and strategies for delivering research during a pandemic are poorly characterised.

Anecdotally, many countries have a baseline shortage of clinical academics in translational science [9] and many leading pathfinder health institutions are within major international transport hubs (London, Madrid, New York), which are affected early in the pandemic. Lockdowns close university departments and funding bodies, with alternative

funding sources (charities, philanthropy) hit by stockmarket falls and competing demand. Frontline remoteness impedes communication of urgency to decision makers, themselves usually selected for process delivery rather than dynamic adaptability. Critical researchers with relevant virology/immunological/intensive care knowledge are drawn in to local or national clinical responses. Other academic staff most likely to redeploy to COVID-19 research self-select for immediate response roles [10] with universities prioritising repurposing to frontline care [11]. High disease rates, required self-isolation periods [12,13] and distractions of remote working degrade the focus needed to create new or repurposed research delivery structures.

We therefore wanted to understand the pandemic research process and describe early lessons. Our aims were to: (i) model potential impact of the pandemic on clinical academic capacity in England relating to COVID-19; and (ii) develop evidence-based recommendations to inform the optimal scientific response to COVID-19.

## Methods

### Cases and excess deaths related to COVID-19 in the UK

Based on our previous analysis of COVID-19 cases and excess deaths in England [14], we considered four scenarios of government interventions associated with different levels of population infection rates: 80% ("*do-nothing*"), 40% ("*relaxed mitigation*"), 10% ("*mitigation*") and 6% ("*Italy model*"), since "partial suppression"(1%) and "full suppression"(0.001%) were no longer feasible. The analyses of excess deaths used data in a cohort design with prospective recording and follow-up from the Clinical reseArch using LInked Bespoke studies and Electronic health Records (CALIBER) open research platform with validated, reusable definitions of several hundred underlying conditions [15,16], linking electronic health records(EHR) from different data sources (via UK unique individual identification data, NHS numbers): primary care (Clinical Practice Research Datalink-GOLD), hospital care (Hospital Episodes Statistics), and death registry (Office of National Statistics). Approval was via the Independent Scientific Advisory Committee (16_022R) of the Medicines and Healthcare products Regulatory Agency in the UK in accordance with the Declaration of Helsinki. Key variables were population infection rate, background mortality risk based on underlying conditions, and relative risk (RR) of mortality associated with COVID-19. We used real-time data until 7 April 2020 for the number of confirmed cases and deaths [17].

### Simulation study for population infection rate and infection growth rate

We designed and implemented a simple stochastic model to predict number of new cases in the population. Since the number of new cases are proportional to the active cases of the previous date (see Web appendix), we used official data from 10 April and calculated the ratio $\frac{\text{new confirmed cases of day n}}{\text{active confirmed cases of day (n−1)}}$ from 5 March onwards. We explored four different scenarios of growth of the infection curve, reflecting different government policies (do-nothing, relaxed mitigation, mitigation and the Italy model), from April 10 (day 36 in our study which coincides with the date of the analysis) until day 250, see Web S1 Table in S1 File.

We assumed that an individual remains infected for 2 weeks, followed by death or immunity, and that actual cases were ~20 times more than confirmed cases, as people with mild or no symptoms are not routinely tested(based on prior estimates of 5- to 100-fold) [18]. Further details are specified in the Web appendix.

## Impact of infection rate on clinical academic workforce

We used NHS Digital data (December 2019) to quantify number of doctors in England (n = 125,119) [19]. Baseline number of clinical academics was estimated as 5% of doctors [20]: 6255 in England. Based on UK clinical academic funding [21], we assumed 50% FTE (Full Time Equivalent) overall, equivalent to ~3000 100% FTE academics, and 25% of doctors off sick and/or socially isolating at any time [22]. There are 1953 intensive care, 7678 emergency medicine, 395 infectious diseases and 2748 respiratory doctors in England [19]. We assumed doctors of any specialty could contribute to the COVID-19 academic response, and necessary research skills and training were homogeneously available throughout the medical workforce.

Clinical academics are not available for research if: (i) they are delivering frontline care due to health system strain, or (ii) they are off sick. We modelled two scenarios with no medical academic capacity at 10% (low strain on the health system) and 5% (high strain on the health system) infection rates respectively. Our outcome was the available number of medical academics during the pandemic in England. We assumed that the number of potentially available 100% FTE clinical academics in research is 3200, but it is obvious that this number has decreased from the early days of the COVID-19 pandemic. We present in detail the assumptions of modelling of available clinical academics in the Web appendix.

## Narrative analysis of case studies

1. Northern Italy: GQ provided first-hand experience of the pandemic as a physician in Bergamo, Italy.

2. Health Care Worker (HCW) cohort study: Our team recently set up and started recruitment for the "Healthcare Worker Bioresource: Immune Protection and Pathogenesis in SARS-CoV-2" Study (COVID-HCW; NCT04318314) [23].

3. Nightingale Hospital: A new NHS field hospital has been established, providing extra medical and intensive care capacity for provision of care to COVID-19 patients with a maximum theoretical capacity of 4000 beds, mainly intensive care [24]. We describe the scenario, staff involved and clinical and research priorities, and constraints.

## Development of recommendations

Based on our model and our case studies, we have developed pragmatic recommendations for clinical research priorities relating to COVID-19.

## Ethical approval

Study approval was granted by the Independent Scientific Advisory Committee (16_022R) of the Medicines and Healthcare products Regulatory Agency in the UK in accordance with the Declaration of Helsinki.

# Results

## Population infection rate and infection growth rate

Fig 1 illustrates different scenarios of infection growth rate and population infection rate (point estimate from x-axis; and cumulative estimate from area under the curve) and the "flatten the curve" phenomenon. The higher the growth rate, the higher the peak infection rate and the quicker the system is overwhelmed by cases of COVID-19. In addition, the course of the

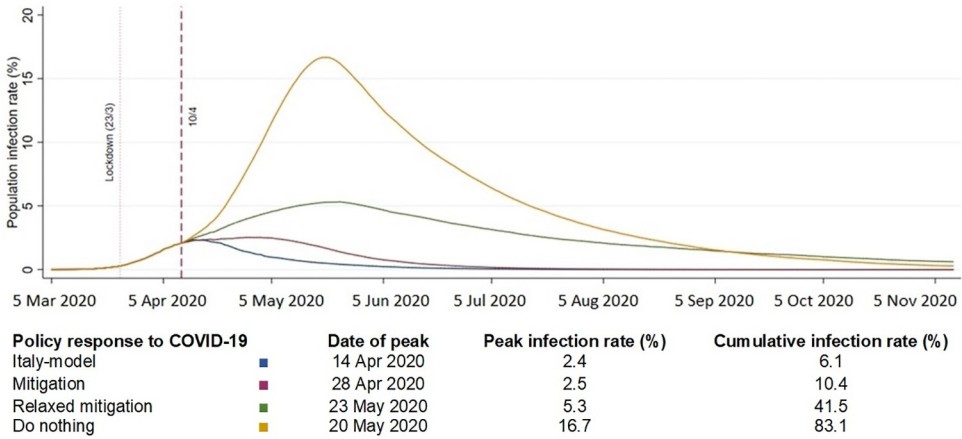

**Fig 1. Population infection rate and daily infection growth rate in the UK during COVID-19 pandemic.**

pandemic is longer. Conversely, if the infection growth rate is reduced, the curve is "flattened" and the pandemic course is shorter. The peak infection rate will be 2.4% (14/4/20), 2.5% (28/4/20), 5.3% (23/5/20) and 16.7% (20/5/20) and the duration of this wave (from 5 March 2020) of the pandemic will be 95, 115, 240 and 240 days respectively. The cumulative infection rates correspond to the scenarios of "do-nothing" (cumulative infection rate ~80%), "relaxed mitigation" (cumulative infection rate ~40%), "mitigation" (cumulative infection rate ~10%) and "Italy model" (cumulative infection rate ~6%).

## Clinical academic capacity

Assuming the "low strain on the health system" model (where there is no academic capacity at population infection rate of 10%), Fig 2 shows that less than 400 100%FTE clinical academics (~13%) will available after April 10 for 11, 34, 62, 76 days for the scenarios of "Italy model", "mitigation", "relaxed mitigation" and "do-nothing" respectively. In the "do nothing scenario",

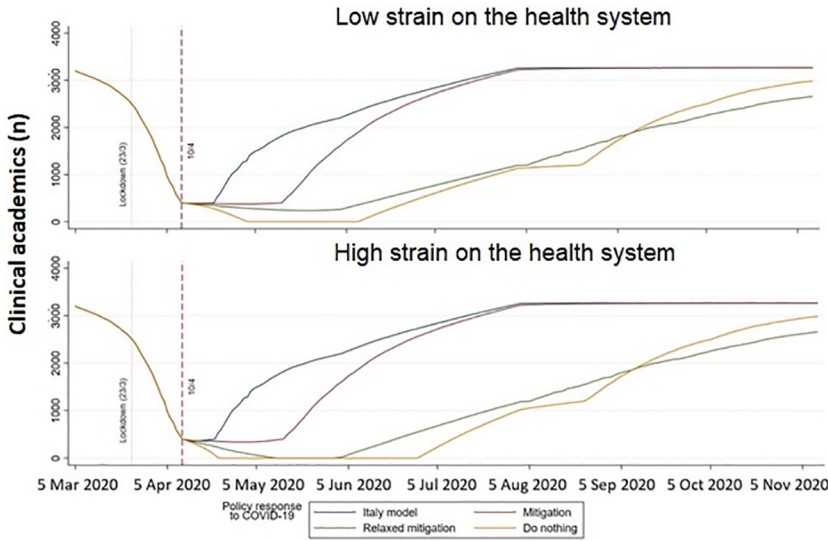

**Fig 2. Clinical academic capacity during the COVID-19 pandemic.**

no clinical academics are available to do research for 37 days (3/5/2020 to 8/6/2020). The predicted dates to reach 2560 clinical academics (80% normal capacity) are 23/6/2020, 1/7/2020, 3/11/2020 and 10/10/2020 for the scenarios of "Italy model", "mitigation", "relaxed mitigation", and "do-nothing", respectively.

In the "high strain on the health system" model (where there is no academic capacity at population infection rate of 5%), in the "relaxed mitigation" scenario, no clinical academics can do research for 18 days (13/5/2020 to 30/5/2020) and in the "do-nothing" scenario, from 23/4/2020 to 28/6/2020. The predicted dates to reach 2560 clinical academics (80% normal capacity) is 23/6/2020, 2/7/2020, 7/11/2020 and 11/10/2020 for the scenarios of "Italy model", "mitigation", "relaxed mitigation", and "do-nothing" respectively.

## Case studies

**Northern Italy.** As the first Western region to be affected (Lombardy, Bergamo) there was effectively no warning. Almost overnight a huge surge of severely ill patients hit us. It was the beginning of a nightmare. With no approved treatments, we had to re-organize the hospital wards, ITU beds, transform simple general into sub-intensive units, commit all doctors from all specialities and research to COVID-19 in a matter of hours-days. It was "catastrophe medicine", research was impossible and approaches were empiric based on analogy to other diseases. Autopsy was our only science. "Mors ubi gaudet succurrere vitae—Where the dead are happy to help the living" and we started to appreciate the high rate of thrombotic complications and pulmonary pathology, initiating empiric anticoagulation and corticosteroids. Only weeks later with external partnerships were formal randomized trials initiated.

**The COVID-HCW study.** Italy informed our strategy. The team nucleus was 5 senior lecturers and 1 professor, all of whom had all their clinical work stopped as irrelevant in the pandemic (cardiac MRI) and who had a track record of monthly large grant writing and detailed systems knowledge. The hospital had no emergency department, so it was protected with a large institution behind it. "Exponential teams" were created to deliver national and local components of research permissions—permissions took 100 documents and ~40 staff working at least part-time to deliver in 7 days (covid-consortium.com). Following scoping, we rejected all but the most basic of aspirations: to capture (questionnaire, bloods and nasal swab) 400 HCW and track changes over 16 weeks—no Clinical Trial of an Investigational Medicinal Product (CTIMP) and no direct work with COVID-19 patients. By day 16 from concept, 400 HCWs had been recruited and the study was in follow-up. At this stage, funding, aliquoting, and detailed basic science plans were embarked on [23].

**Nightingale hospital.** The Nightingale Hospital was the largest field hospital in Europe with the largest number of intensive care and step-down facilities for COVID-19. It was set up in 14 days from initial concept to first patient admitted Nightingale is a learning system, underpinned by research. For patients, staff and wider NHS benefit, the design incorporates a commitment to learning fast and acting fast across all dimensions: clinical, operational, and staff wellbeing. Our research approach is: (i) embedded within the Quality and Learning team, (ii) simple; and (iii) high-quality and high-volume recruitment. The onsite team is backed up by QMUL, UCL and UCLP, with multidisciplinary expertise, including virology, immunology and therapeutics. From an initial two clinical academic staff (AB and JM), a research governance structure has been set up rapidly and a simple strategy has been established. COVID-19 consented studies can be observational or interventional (drugs), in patients or staff. We plan just one initial study in each domain, choosing the simplest possible approaches: patient observational(ISARIC), patient therapeutic(RECOVERY), staff observational (COVID-HCW with expansion to n = 1000) and staff therapy (pre- and/or post-exposure prophylaxis studies-to be

**Table 1. Simple strategy for consented studies at the Nightingale hospital, London.**

|  | Patients | Staff |
|---|---|---|
| **Observational** | ISARIC www.remapcap.org/ | HealthCare Workers study www.covid-consortium.org |
| **Randomised trial** | Recovery www.recoverytrial.net/ (later REMAP-CAP) www.remapcap.org/ | (a pre-exposure prophylaxis study– 3 in preparation: we will choose one) |

**Table 2. Research strategy recommendations.**

1. **Radical prioritisation**: Few research ideas, e.g. 3–4 per global institution
2. **Leadership**: ideally from those without immediately transferrable clinical skills.
3. **Rationalisation**:
   a. Data analysis using existing systems
   b. Simple delivery of samples
4. **Resourcing**: Profound project resourcing to deliver—essentially exponentially more staff than usual (e.g. estimate of up to 50 people per project immediately).
5. **Careful site selection**:
   a. Clinical research: in the community or in large academic health centres
   b. Basic science: using teams without clinical or basic science transferable skills for COVID-19 work, and protecting research workers from clinical service duties.
6. **Dissemination**: across UK and internationally to those considering COVID-19 clinical research. Data sharing and collaboration both nationally and internationally.

confirmed]). The first patient therapeutic trial patient will be recruited on day 8 after first patient admitted. Other studies (data, staff surveys) can be conducted at other sites (Table 1) or after the initial exponential wave peaks. In addition, there are opportunities for non-consented research, such as epidemiologic and advanced data analytics; e.g. initiatives such as DeCOVID [25] to mobilise data, computer scientists, analysts and analytic infrastructure, including and clinical expertise. There is potential to link effective learning directly to and from clinical questions.

## Recommendations

After discussion among co-authors, and consensus among a stakeholder group at the Nightingale Hospital, we produced recommendations for a COVID-19 clinical research strategy (Table 2).

## Discussion

In the first study of clinical academic capacity in the COVID-19 era, we show the existential threat to research responses facing the UK and other countries. Urgent recognition and mobilisation are required to ensure prioritisation of the most appropriate and clinically imperative science. We have developed recommendations relevant to all health systems.

The healthcare and public health emergency caused by COVID-19 is not in question, fuelling global discussion, modelling and multidisciplinary research at a pace rarely seen [26,27]. However, strain on clinical academic workforce and infrastructure in different countries are notable omissions. Despite programmes to promote research preparedness in epidemics, COVID-19 poses particular challenges [28] to our responses which rest ultimately on research, whether vaccines, drugs, ventilation strategies, risk prediction or machine learning. Our experiences are echoed in China, Italy and other countries facing the pandemic.

There have been quick efforts and advances in fields as diverse as genomics [29] and data science [30], with rapid-response calls from major funders [5,31–33]. However, our data signal a need for a far broader paradigm shift in research design and implementation. At every stage in the traditional research pipeline, there are roadblocks hampering swift reactions necessary to tackle COVID-19 within and across countries. Even on a "war footing", research processes are unnecessarily time- and resource- consuming, particularly when involving randomized controlled studies. Specific hurdles are: (i) **Staff** -doctors and research nurses, but also access to labs; (ii) **Stuff**-consumables difficult to obtain due to challenging supply chains especially if they are competing with clinical service delivery, e.g. personal protective equipment; (iii) **Site**- ideally research space near to clinical areas; and **Systems**- approvals in a timely fashion, e.g. Research Ethics Committee, Health Research Authority, Local Research and Development team and standard operating procedures ins relevant institutions.

Emergencies as far-reaching as the current scenario require total rethinking of research delivery, and aspects that work better when some of the processes are accelerated and the permissions expedited, may well yield long-term benefits outside of COVID-19 research. Here we have modelled clinical academic time in terms of numbers of staff and time in the pandemic. However, a far deeper examination of the role of clinical academics beyond "hours at the desk" is warranted in times of public health emergency to include the "what" and "how" of their work. For example, certain tasks such as research permissions and data analysis may be diverted away from clinical academics, who may be better placed to act as conduits between the clinical and public health spheres and teams of non-clinical researchers. The needs of the hour are patient-centred, data-driven and time-responsive, and it may be time to usefully change the role and function of the clinical academic. It is worth noting that this is occurring against a backdrop of declining clinical academic numbers [21].

Our simulations suggest the pandemic will create health system strain for many critical months. Depending on a range of COVID-related factors, we show that the clinical academic workforce may be depleted when it is needed most to lead and conduct clinical research, even in a relatively well-resourced context such as the UK, whether by funding, number of universities and staff, infrastructure or policy. Therefore, other countries are likely to be worse affected. COVID-19 research is least likely to occur where it is most needed, magnifying the well-documented "10–90 research gap", where only 10% of resources for global healthcare research are devoted to low-income settings where 90% of preventable deaths occur [34]. Although COVID-19 is a unique threat, there are lessons to be learned from prior health research strategies to address structural inequities, such as the Global Fund for Malaria, TB and HIV/AIDS [29]. Without coordinated international responses, including urgent funding and infrastructure, research will be retrospective, patchy and unlikely to have an effect.

We provide six clear recommendations for science in the UK and globally in relation to COVID-19 (Radical prioritisation, Leadership, Rationalisation, Resourcing, Careful site selection and Dissemination). Radical prioritisation is important where field hospitals are being established in rapid timescales in different countries with delivery constraints. High-quality evidence can be obtained, but studies need to be lean with minimal complexity for key operational steps: consenting, randomisation, drug delivery, monitoring, outcomes and follow-up. The number of patients recruited to deliver definitive answers needs to be large, with fast recruitment across multiple sites. Furthermore, adaptive trial designs are preferred as new arms (e.g. multi-drug) can be generated swiftly and other arms dropped (e.g. supportive care if one arm has a signal of efficacy) without restarting permissions, via substantial amendments [35,36].

Leadership and rationalisation are the next key steps. Balance needs to be struck between clinical researchers in contact with the "frontline" so that research questions are clinically

relevant and timely, and having research active leaders who will not be protected from front-line work. Rationalisation involves a study selection strategy that is deeply resourced for a limited number (1 or 2) studies per COVID cohort. In selecting these studies, a single study of one investigational medical product versus standard of care (supportive care) with 50:50 randomisation is inefficient compared to studies with multiple therapeutic arms. Most single agent approaches to COVID-19 are likely to have, at most, a modest effect.

We used a stochastic model accounting for infection rate, infection growth rate and clinical academic capacity using up-to-date official statistics. There are limitations to our model and its assumptions. Our model was simple and was based only on observational patterns of the number of new cases and actual cases from publicly available data [17]. We conducted analyses on 10 April, and on 11 April, some extra ~3000 cases were added retrospectively and distributed over the past 10 days-we did not include these data. It did not take into account infectious disease epidemiology parameters, such as the basic reproductive number (R0), and we did not consider differing levels of risk of infection [37–39]. Our model on the availability of clinical academics makes several assumptions (Web Appendix), including the total number of 100% FTE academics as ~3200, with a uniform skillset across the workforce.

## Conclusions

In the first study to model and estimate the impact of COVID-19 on the coordinated clinical academic response at system level, we show that all countries face depletion of their clinical academic workforce for several months, which will greatly hamper research in prevention and treatment. The number of studies needs to be rationalised urgently and background problems in clinical academia need to be overcome quickly. To quote Sir Jeremy Farrar, "The only exit from this pandemic is through science" [40] and that requires staffing.

## Supporting information

**S1 File. Web appendix.**
(DOCX)

**S2 File. Worldometer UK.**
(XLSX)

**S3 File. Worldometer Italy.**
(XLSX)

**S4 File. Available doctors in the UK.**
(XLSX)

## Author Contributions

**Conceptualization:** Amitava Banerjee, James C. Moon.

**Data curation:** Amitava Banerjee, Michail Katsoulis.

**Formal analysis:** Amitava Banerjee, Michail Katsoulis.

**Investigation:** Amitava Banerjee.

**Methodology:** Amitava Banerjee, Michail Katsoulis.

**Project administration:** Amitava Banerjee.

**Resources:** Amitava Banerjee, Michail Katsoulis.

**Supervision:** Amitava Banerjee.

**Visualization:** Michail Katsoulis, Alvina G. Lai.

**Writing – original draft:** Amitava Banerjee, James C. Moon.

**Writing – review & editing:** Amitava Banerjee, Michail Katsoulis, Alvina G. Lai, Laura Pasea, Thomas A. Treibel, Charlotte Manisty, Spiros Denaxas, Giovanni Quarta, Harry Hemingway, João L. Cavalcante, Mahdad Noursadeghi, James C. Moon.

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
