## [Decision Letter · Decision Letter 0]

9 Jun 2020

PONE-D-20-11840

Clinical academic research in the time of Corona: a simulation study in England and a call for action

PLOS ONE

Dear Dr. Banerjee,

Thank you for submitting your manuscript to PLOS ONE. After careful consideration, we feel that it has merit but does not fully meet PLOS ONE’s publication criteria as it currently stands. Therefore, we invite you to submit a revised version of the manuscript that addresses the points raised during the review process.

We look forward to receiving your revised manuscript.

Kind regards,

Marco Remondino

Academic Editor

PLOS ONE

Journal Requirements:

AB is supported by research funding from NIHR, British Medical Association, Astra-

Zeneca and UK Research and Innovation. BW and HH are National Institute for Health

Research (NIHR) Senior Investigators and are funded by the National Institute for Health

Research University College London Hospitals Biomedical Research Centre. HH work is

supported by: 1. Health Data Research UK (grant No. LOND1), which is funded by the UK

Medical Research Council, Engineering and Physical Sciences Research Council, Economic

and Social Research Council, Department of Health and Social Care (England), Chief Scientist

Office of the Scottish Government Health and Social Care Directorates, Health and Social

Care Research and Development Division (Welsh Government), Public Health Agency

(Northern Ireland), British Heart Foundation and Wellcome Trust. 2. The BigData@Heart

Consortium, funded by the Innovative Medicines Initiative-2 Joint Undertaking under grant

agreement No. 116074. This Joint Undertaking receives support from the European Union’s

Horizon 2020 research and innovation programme and EFPIA; it is chaired, by DE Grobbee

and SD Anker, partnering with 20 academic and industry partners and ESC. This work was

supported by a National Institute of Health Research (NIHR) Clinician Scientist award (CS-

2016-007) to L.S.

No specific funding was received for this work.

5. We note that Figure 3 in your submission contain copyrighted images. All PLOS content is published under the Creative Commons Attribution License (CC BY 4.0), which means that the manuscript, images, and Supporting Information files will be freely available online, and any third party is permitted to access, download, copy, distribute, and use these materials in any way, even commercially, with proper attribution. For more information, see our copyright guidelines: http://journals.plos.org/plosone/s/licenses-and-copyright.

a. You may seek permission from the original copyright holder of Figure 3 to publish the content specifically under the CC BY 4.0 license.

Additional Editor Comments (if provided):

Please read carefully the comments below and address the few issues.

Reviewers' comments:

Reviewer's Responses to Questions

**Comments to the Author**

1. Is the manuscript technically sound, and do the data support the conclusions?

Reviewer #1: Yes

2. Has the statistical analysis been performed appropriately and rigorously? 

Reviewer #1: Yes

3. Have the authors made all data underlying the findings in their manuscript fully available?

Reviewer #1: Yes

4. Is the manuscript presented in an intelligible fashion and written in standard English?

Reviewer #1: Yes

5. Review Comments to the Author

Reviewer #1: the manuscript "Clinical academic research in the time of Corona: a simulation study in England and a call for action" is an important topic given the ongoing COVID-19 pandemic and the results are interesting to communicate. The manuscript is well written and the authors used a standardized methodology. The abstract appropriately cover the contents of the article and the presentation reflect the present state of knowledge and the literature is sufficiently critical, current, and internationally evaluated. The text presented and arranged clearly and concisely and the conclusions been justified sufficiently. The English language is standard

6. PLOS authors have the option to publish the peer review history of their article (what does this mean?). If published, this will include your full peer review and any attached files.

Reviewer #1: Yes: Ekram Wassim Abd El-Wahab

---

## [Author Response · Author response to Decision Letter 0]

9 Jun 2020

Thanks-we have done this.

Thanks-we have followed this instruction.

Thanks-now addressed.

It is not necessary to adjust the disclosure-thanks.

We look forward to receiving your revised manuscript.

Kind regards,

Marco Remondino

Academic Editor

PLOS ONE

Journal Requirements:

We have adjusted to meet all the stylistic requirements.

AB is supported by research funding from NIHR, British Medical Association, Astra-

Zeneca and UK Research and Innovation. BW and HH are National Institute for Health

Research (NIHR) Senior Investigators and are funded by the National Institute for Health

Research University College London Hospitals Biomedical Research Centre. HH work is

supported by: 1. Health Data Research UK (grant No. LOND1), which is funded by the UK

Medical Research Council, Engineering and Physical Sciences Research Council, Economic

and Social Research Council, Department of Health and Social Care (England), Chief Scientist

Office of the Scottish Government Health and Social Care Directorates, Health and Social

Care Research and Development Division (Welsh Government), Public Health Agency

(Northern Ireland), British Heart Foundation and Wellcome Trust. 2. The BigData@Heart

Consortium, funded by the Innovative Medicines Initiative-2 Joint Undertaking under grant

agreement No. 116074. This Joint Undertaking receives support from the European Union’s

Horizon 2020 research and innovation programme and EFPIA; it is chaired, by DE Grobbee

and SD Anker, partnering with 20 academic and industry partners and ESC. This work was

supported by a National Institute of Health Research (NIHR) Clinician Scientist award (CS-

2016-007) to L.S.

No specific funding was received for this work.

 Apologies for this oversight. We have now removed the funding section from the manuscript.

 This is now amended-apologies for the oversight.

 This has now been amended in the revised manuscript.

5. We note that Figure 3 in your submission contain copyrighted images. All PLOS content is published under the Creative Commons Attribution License (CC BY 4.0), which means that the manuscript, images, and Supporting Information files will be freely available online, and any third party is permitted to access, download, copy, distribute, and use these materials in any way, even commercially, with proper attribution. For more information, see our copyright guidelines: http://journals.plos.org/plosone/s/licenses-and-copyright.

a. You may seek permission from the original copyright holder of Figure 3 to publish the content specifically under the CC BY 4.0 license.

Thanks for this information. We have now removed figure 3 and any reference to it in the revised manuscript.

Additional Editor Comments (if provided):

Please read carefully the comments below and address the few issues.

Reviewers' comments:

Reviewer's Responses to Questions

Comments to the Author

1. Is the manuscript technically sound, and do the data support the conclusions?

Reviewer #1: Yes

2. Has the statistical analysis been performed appropriately and rigorously? 

Reviewer #1: Yes

3. Have the authors made all data underlying the findings in their manuscript fully available?

Reviewer #1: Yes

4. Is the manuscript presented in an intelligible fashion and written in standard English?

Reviewer #1: Yes

5. Review Comments to the Author

Reviewer #1: the manuscript "Clinical academic research in the time of Corona: a simulation study in England and a call for action" is an important topic given the ongoing COVID-19 pandemic and the results are interesting to communicate. The manuscript is well written and the authors used a standardized methodology. The abstract appropriately cover the contents of the article and the presentation reflect the present state of knowledge and the literature is sufficiently critical, current, and internationally evaluated. The text presented and arranged clearly and concisely and the conclusions been justified sufficiently. The English language is standard

 We thanks the reviewer for these kind remarks.

6. PLOS authors have the option to publish the peer review history of their article (what does this mean?). If published, this will include your full peer review and any attached files.

Do you want your identity to be public for this peer review? For information about this choice, including consent withdrawal, please see our Privacy Policy.

Reviewer #1: Yes: Ekram Wassim Abd El-Wahab

This is now done.

---

## [Decision Letter · Decision Letter 1]

27 Jul 2020

Clinical academic research in the time of Corona: a simulation study in England and a call for action

PONE-D-20-11840R1

Dear Dr. Amitava Banerjee,

We’re pleased to inform you that your manuscript has been judged scientifically suitable for publication and will be formally accepted for publication once it meets all outstanding technical requirements.

Kind regards,

Francesco Di Gennaro

Academic Editor

PLOS ONE

Additional Editor Comments (optional):

I read and evaluete your manuscript with great interest.

I find it well wrote and very high quality

I suggest to accept it

Congratulations

Reviewers' comments:

Reviewer's Responses to Questions

**Comments to the Author**

1. If the authors have adequately addressed your comments raised in a previous round of review and you feel that this manuscript is now acceptable for publication, you may indicate that here to bypass the “Comments to the Author” section, enter your conflict of interest statement in the “Confidential to Editor” section, and submit your "Accept" recommendation.

Reviewer #1: All comments have been addressed

2. Is the manuscript technically sound, and do the data support the conclusions?

Reviewer #1: Yes

3. Has the statistical analysis been performed appropriately and rigorously? 

Reviewer #1: Yes

4. Have the authors made all data underlying the findings in their manuscript fully available?

Reviewer #1: Yes

5. Is the manuscript presented in an intelligible fashion and written in standard English?

Reviewer #1: Yes

6. Review Comments to the Author

Reviewer #1: The author have properly addressed all the comments and the manuscript is suitable for publication.

7. PLOS authors have the option to publish the peer review history of their article (what does this mean?). If published, this will include your full peer review and any attached files.

Reviewer #1: **Yes: **Ekram Wassim Abd El-Wahab

---

## [Editor Report · Acceptance letter]

3 Aug 2020

PONE-D-20-11840R1 

Clinical academic research in the time of Corona: a simulation study in England and a call for action 

Dear Dr. Banerjee:

I'm pleased to inform you that your manuscript has been deemed suitable for publication in PLOS ONE. Congratulations! Your manuscript is now with our production department. 

Kind regards, 

on behalf of

Dr. Francesco Di Gennaro 

Academic Editor

PLOS ONE